# Robust Control Design of a Human Heart Rate System for Cardiac Rehabilitation Exercise

Saad Jamshed Abbasi [1], Won Jae Kim [2], Jaehyung Kim [2], Min Cheol Lee [2,*], Byeong Ju Lee [3] and Myung Jun Shin [3]

1    School of Aerospace and Mechanical Engineering, Korea Aerospace University,
     Goyang 10540, Republic of Korea
2    School of Mechanical Engineering, Pusan National University, Busan 46241, Republic of Korea
3    Department of Rehabilitation Medicine, Pusan National University Hospital, Busan 49241, Republic of Korea
*    Correspondence: mclee@pusan.ac.kr

**Abstract:** Automatic, precise, and accurate heart rate control during treadmill exercise is an interesting topic among researchers. The human heart is a highly nonlinear system. Conventional control techniques are not sufficient and it is difficult to accurately model the human heart. Two different robust controllers were designed for this nonlinear system. Firstly, sliding mode control (SMC) was implemented; SMC is robust against parametric uncertainties and external disturbance but its robustness is not guaranteed during the reaching phase, especially in heart rate control, and implementation of SMC requires the linear parameters of the system (human heart rate model). In this research, the signal compression method (SCM) was used for approximately linearized modeling of the human heart rate. The extraction of the human heart rate model using SCM requires experiment and computation. Furthermore, it was observed in this research that SCM is not a precise method. Therefore, integral sliding mode control (ISMC) was designed and implemented to overcome these difficulties. By introducing an auxiliary sliding surface, the reaching phase and effect of the perturbation on an actual sliding surface were eliminated; furthermore, implementation of ISMC does not require the linear parameters of the system. Simulations were performed in MATLAB/Simulink and experiments were conducted in a hospital. Six clinical subjects participated in this experiment. Both forms of control logic were implemented during the desired heart rate tracking test. Results showed that the desired heart rate tracking of ISMC is better than that of SMC. The tracking error of ISMC is smaller than that of SMC. However, ISMC control output has chattering, which needs to be reduced.

**Keywords:** auxiliary sliding surface; actual sliding surface; integral sliding mode control; perturbation; sliding mode control

## 1. Introduction

In recent years, mortality caused by heart issues and disease has increased in South Korea. Different advanced countries are developing community health care departments to predict disease and health issues in advance to reduce illness rates. Similarly, statistical analysis of different diseases and their corresponding effects provides a generic overview of areas that require human attention. The recent statistical data indicate that cancer, followed by heart disease, are major causes of death in South Korea [1].

The number of patients with heart disease has steadily increased. The rehabilitation of heart patients is important for their life. Currently, there are different heart rehabilitation centers all over the world. The main function of these centers is, through exercise, to improve the physical health of patients, which has been damaged by heart disease. These exercises include walking, running, cycling, and swimming. Rehabilitation centers provide treadmills for running exercises. In South Korea, there are 11 different centers for heart

patient rehabilitation, in which trained manpower is limited compared with the centers in developed countries. These centers use manual techniques to achieve the desired heart rate (prescribed by the doctor) during treadmill exercise. Due to constraints on operations, these exercises are not safe for heart patients. Therefore, it is required that heart rates are controlled by some automated algorithm to ensure safe implementation.

The human heart is a highly nonlinear system and it is a difficult task to develop a precise model. In the past, different researchers have proposed unique techniques to extract a human heart model, each having advantages and disadvantages [2–12]. Yalcinkaya et al. [13] proposed a human heart model by considering it as a hydro-electromechanical system. They simulated the human heart based on three main functions: hydraulic, electrical, and mechanical parameters. The developed hydro-mechanical system was transformed into an electrical domain. The corresponding simulation was carried out according to the mathematical model or formulations obtained using the Laplace transform. Cheng et al. [14] proposed a unique idea to obtain a nonlinear model of the human heart through feedback interconnected systems. Recently, researchers have undertaken mathematical modeling using artificial intelligence techniques, but these require more time to achieve a model of the human heart rate. Li et al. [15] proposed a novel method to obtain a human heart rate model with a reduced time span. They used the signal compression method (SCM) to obtain a human heart rate model. In this research, SCM was used to extract an approximately linearized human heart rate model [16].

The human heart is a nonlinear system and the design of a controller to track the reference heart rate during treadmill exercise is a challenging task. Thus, researchers have proposed different control algorithms to achieve this task [17–20]. Su et al. [18] proposed a fuzzy neural network to control the treadmill speed during exercise. Cheng et al. [14] proposed feedforward and feedback techniques to track the desired heart rate trajectory. Su et al. proposed another method based on H infinity control approaches for human heart rate control during treadmill exercise [19]. It is not easy to implement the above-mentioned control techniques because of their computational complexities. Kim et al. [17] presented linear control (PI) for human heart rate control during exercise. It is easy to implement PI control but the heart rate tracking error is large. Because the heart rate system is nonlinear, a robust control algorithm is needed to reduce output error and ensure stability.

In this research, robust controllers were designed for human heart rate control during treadmill exercise. Initially, sliding mode control (SMC) was derived for a specific dynamical system. SMC [21] is a variable structure control that utilizes a switching control law to alter the plant dynamics such that the plant states slide along the sliding surface [22–32]. SMC has two phases, which are the reaching phase and the sliding phase. In the reaching phase, the plant states are forced to move towards the sliding surface with the help of the switching gain as the system states reach the sliding surface. In the next phase, which is known as the sliding phase, they slide along the origin. In the sliding phase, the system remains insensitive to uncertainties and external disturbances. However, during the reaching phase, robustness of SMC is not guaranteed. In SMC, actual sliding surface dynamics are affected by perturbation [33]. In addition, the design of SMC requires linear parameters of the system (human heart rate model). In this research, SCM was used to extract a linear dynamic model from the human heart rate one. Experiment and computation are required to extract the linearized human heart rate model using SCM. Therefore, this research requires two different experiments: the first is to extract the human heart rate model (system identification experiment), and the second is to track the desired heart rate trajectory (control experiment). To overcome the above-mentioned problems, integral sliding mode control (ISMC) was implemented [33]. An auxiliary sliding variable in ISMC was designed by introducing the integral term in the conventional sliding surface. This auxiliary sliding variable can eliminate the reaching phase [34,35]. Therefore, actual sliding surface dynamics have no perturbation effect. The controller design of ISMC does not require the human heart rate model. The control inputs in ISMC are divided into two parts. The first part compensates for the perturbation plus system dynamics, and the second part

forces the system state to move towards the origin in a satisfactory time. The designed controllers were implemented in MATLAB for simulation purposes and later applied to the real system, and corresponding data and results were obtained. The experiments were conducted on clinical subjects after receiving the permission of the institutional review board (IRB # H-1904-016-077) at Pusan National University Hospital, Busan, South Korea. These subjects are healthy but they have greater chance of heart disease in the future. The results show that ISMC performance is better than that of SMC. In this study, the main focus was to design a robust controller for desired human heart rate tracking. A brief review of human heart rate modeling is presented in this manuscript based on the mathematical model designed by Li and Lee [15].

This manuscript is organized as follows: Section 2 presents the experimental details and mathematical formulation of the human heart rate model. Section 3 presents the proposed control methods for human heart rate control. Section 4 provides the simulation details, experimental results, and comparisons. Section 5 comprises the concluding remarks of this study.

## 2. Experimental Setup and Human Heart Rate Modeling

This section presents the experimental setup and human heart rate modeling.

### 2.1. Experimental Setup

A HERA 9000 model treadmill exercise machine manufactured by HEALTH ONE CO, LTD, Korea, was used in the experiment. The maximum speed of the treadmill is 20 km/h and inclination can be adjusted up to 16 percent. A heart rate monitor sensor is equipped on the left arm of the treadmill, and discreetly measures the heart rate of the clinical subject. Additionally, the control panel of the treadmill allows for the user to input the desired heart rate and the inclination of the platform. The designed treadmill controller in this study can control the desired heart rate. It was observed during the experiment that holding the handle is difficult for the clinical subjects due to sweating. Furthermore, sweating also caused noise in the heart rate measurement sensor installed on the treadmill handle. Thus, an additional sensor for heart rate measurement was attached to the index finger and wrist of the clinical subject with the permission of the hospital advisory committee. There is a stop button in the treadmill control panel in case of an emergency.

The sensor was connected with a controlled PC through Bluetooth and the data were discreetly saved for further processing. Furthermore, the treadmill system was exogenously controlled through the PC, and the corresponding diagram is shown in Figure 1. This connection was made through a communication wire.

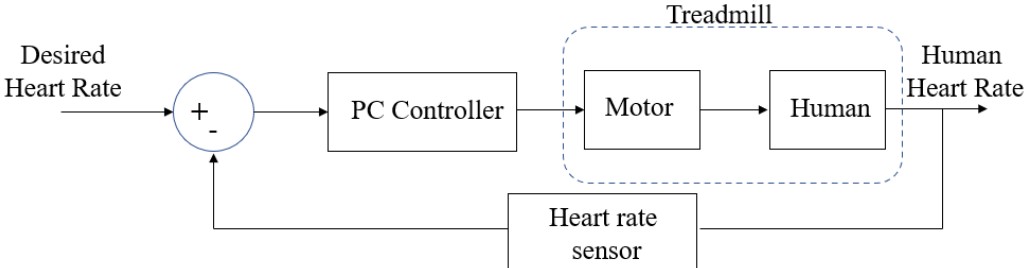

**Figure 1.** Experimental setup block diagram.

The heart rate measurement sensor is a NONIN 350, which measures the heart rate and oxygen saturation level in the blood. A graphical user interface (GUI) was designed in the control PC, and was implemented in C#. This GUI was used to visualize results and analyze data, and had the capability of input by the user. The visual display of the GUI showed the treadmill velocity, pulse oximetry data, input/output parameters, and the stop/run command block, as shown in Figure 2.

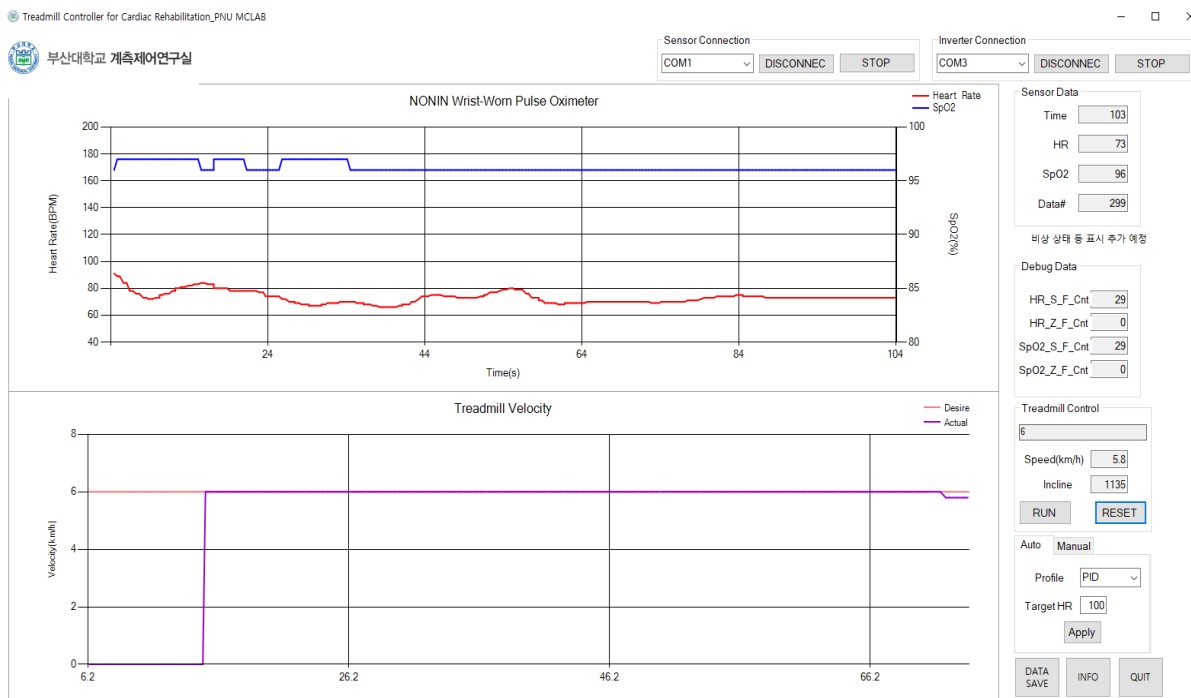

**Figure 2.** GUI generated using Microsoft Visual Studio. (Korean: Pusan National University, Measurement and Control Lab).

### 2.2. Human Heart Rate Modeling

In this research, the heart rate models of six different Korean clinical subjects were extracted using the signal compression method (SCM) [15]. The experiments were conducted in Pusan National University Hospital, Busan, South Korea to observe/measure the required data for heart modeling. The data corresponding to the cardiopulmonary exercise testing (CPET) of each subject are displayed in Table 1 and were provided by the hospital. Currently, the subjects are healthy but they have greater chance of heart disease in the future. The estimated linear term's parameter of the human heart rate model can be derived in Table 2. The corresponding second-order heart rate model of each subject is assumed and derived through a general second-order system (1):

$$G = \frac{\omega_n^2}{s^2 + 2 \cdot \delta \cdot \omega_n \cdot s + \omega_n^2}, \tag{1}$$

where $\zeta$ is the damping ratio and $\omega_n$ is the natural frequency of the system.

**Table 1.** Data of clinical subjects who participated in the experiment.

| Clinical Subject | Data | | | | |
|---|---|---|---|---|---|
| | Age | Gender | Rest Heart Rate | Maximum Heart Rate | Decay Heart Rate |
| A | 54 | F | 88 | 190 | 23 |
| B | 61 | M | 59 | 151 | 24 |
| C | 28 | F | 74 | 172 | 36 |
| D | 41 | F | 73 | 152 | 8 |
| E | 54 | F | 70 | 152 | 20 |
| F | 62 | F | 73 | 149 | 17 |

**Table 2.** Estimated heart rate model parameter using SCM.

| Clinical Subject | Subject Information | | |
|:---:|:---:|:---:|:---:|
| | $\delta$ | $\omega_n$ | Correlation (%) |
| A | 4.3 | 0.5 | 82 |
| B | 4.5 | 0.5 | 86 |
| C | 5.7 | 0.6 | 83 |
| D | 4.3 | 0.5 | 85 |
| E | 7 | 0.6 | 84 |
| F | 5.6 | 0.6 | 79 |

## 3. Controller Design

This section presents the controller design; SMC and then ISMC are discussed. The performance characteristics of ISMC are presented in detail.

### 3.1. Sliding Mode Control

A sliding mode control (SMC) is a variable structure control that is used for nonlinear systems due to its invariance to both parametric uncertainties and external disturbances. In SMC, the main idea is to design the sliding surface ($\sigma$):

$$\sigma = \dot{e} + c \cdot e, \tag{2}$$

where $c$ is the constant and $e$ is the error between the actual and desired value ($e = x_d - x$). SMC consists of two different control inputs:

$$u = u_{sw} + u_{eq}, \tag{3}$$

where $u_{sw}$ forces the system state to reach towards the designed sliding surface. Once the error reaches the sliding surface, then it converges to zero. $u_{eq}$ keeps the error at the sliding surface. There are two phases in SMC: the reaching phase and the sliding phase. During the reaching phase, the switching control forces the system state (initial error) to move towards the sliding surface in finite time. Once the error reaches the sliding surface, then it moves towards the origin (zero). A general second-order system including disturbances is represented in (4):

$$\ddot{x} + f(x, \dot{x}, t) + \Delta f(x, \dot{x}, t) + d = u, \tag{4}$$

where $u$ is the control input, $x$ represents the system states, $f(x, \dot{x}, t)$ represents the system linear dynamics, $\Delta f(x, \dot{x}, t)$ represents the uncertainties in dynamics, and $d$ is the external disturbance. The controller task is to move the system states to a desired value in a satisfactory time. System uncertainties and external disturbance are considered as a perturbation in (5), and (6) is derived using (4) and (5):

$$\Psi(x, \dot{x}, t) = \Delta f(x, \dot{x}, t) + d, \tag{5}$$

$$\ddot{x} = u - f(x, \dot{x}, t) - \Psi(x, \dot{x}, t). \tag{6}$$

To reach the sliding surface ($\sigma = 0$), the Lyapunov stability criterion should be satisfied:

$$\sigma \cdot \dot{\sigma} \leq 0. \tag{7}$$

To satisfy Lyapunov stability:

$$\dot{\sigma} = -K\text{sat}(\sigma), \tag{8}$$

where $K$ is the switching gain and $\text{sat}(\sigma)$ is the saturation function, defined as:

$$\text{sat}(\sigma) = \begin{cases} \sigma/|\sigma|, & \text{if } |\sigma| > \varepsilon_{cs} \\ \sigma/\varepsilon_{cs}, & \text{if } |\sigma| \leq \varepsilon_{cs} \end{cases}, \tag{9}$$

where $\epsilon_c$ is the boundary layer thickness. The control input can be derived using (2), (6), and (8):

$$u = K\text{sat}(\sigma) + f(x, \dot{x}, t) + c \cdot \dot{e} + \ddot{x}_d, \tag{10}$$

$$u_{sw} = K\text{sat}(\sigma), \tag{11}$$

$$u_{eq} = f(x, \dot{x}, t) + c \cdot \dot{e} + \ddot{x}_d. \tag{12}$$

To achieve finite-time convergence, the magnitude of the switching gain ($K$) should be greater than the upper bound of perturbation:

$$K > |\Psi|. \tag{13}$$

**Theorem 1.** *For the second-order system described by (4), the Lyapunov stability criterion (7) can be achieved under the condition (13).*

**Proof.** It must be noted that for the system described by (4) with proposed control law (10), the system stability (7) can be achieved by enforcing the condition (13):

$$\sigma \cdot \dot{\sigma} \leq \sigma(\ddot{e} + c \cdot \dot{e}) \leq 0, \tag{14}$$

Using (2), (6) and (10), (14) can be rewritten as:

$$\sigma \cdot \dot{\sigma} \leq \sigma(\ddot{x}_d - K\text{sat}(\sigma) - f(x, \dot{x}, t) - c \cdot \dot{e} - \ddot{x}_d + f(x, \dot{x}, t) + \Psi(x, \dot{x}, t) + c \cdot \dot{e}) \leq 0, \tag{15}$$

After solving the above relation:

$$\sigma \cdot \dot{\sigma} \leq \sigma(-K\text{sat}(\sigma) + \Psi(x, \dot{x}, t)) \leq 0. \tag{16}$$

To keep the system stable, the gain $K$ should be greater than the absolute magnitude of perturbation ($\Psi$). The sliding surface dynamics during the reaching phase can be calculated as:

$$\dot{\sigma} = -K\text{sat}(\sigma) + \Psi(x, \dot{x}, t). \tag{17}$$

As can be observed in (17), the sliding surface dynamics are affected by the perturbation (when $\sigma \neq 0$). Therefore, the system stability is not guaranteed during the reaching phase. The drawbacks of the SMC in the desired heart rate control application are discussed in the following. □

### 3.1.1. Sensitive to Perturbation

As can be observed in Figure 3a, during the reaching phase, the robustness of SMC is not guaranteed due to effects of perturbation on sliding surface dynamics ($\dot{\sigma} = -K\text{sat}(\sigma) + \Psi$), if $K < |\Psi|$. This perturbation is the sum of parametric uncertainties, nonlinearity, and external disturbances. This means that the stability of the system is not guaranteed during the reaching phase because the system stability can be only achieved by satisfying the stability condition (13). It is very difficult to model the human heart accurately. In this paper, the technique used to model the human heart was developed by Li et al. [15]. It is worth mentioning that this technique has a limitation because it can only estimate linear parameters. It is not possible to estimate the nonlinearity and parametric uncertainties; therefore, in this study, ISMC was proposed to eliminate the effect of the perturbation on $\dot{\sigma}$ dynamics during the reaching phase.

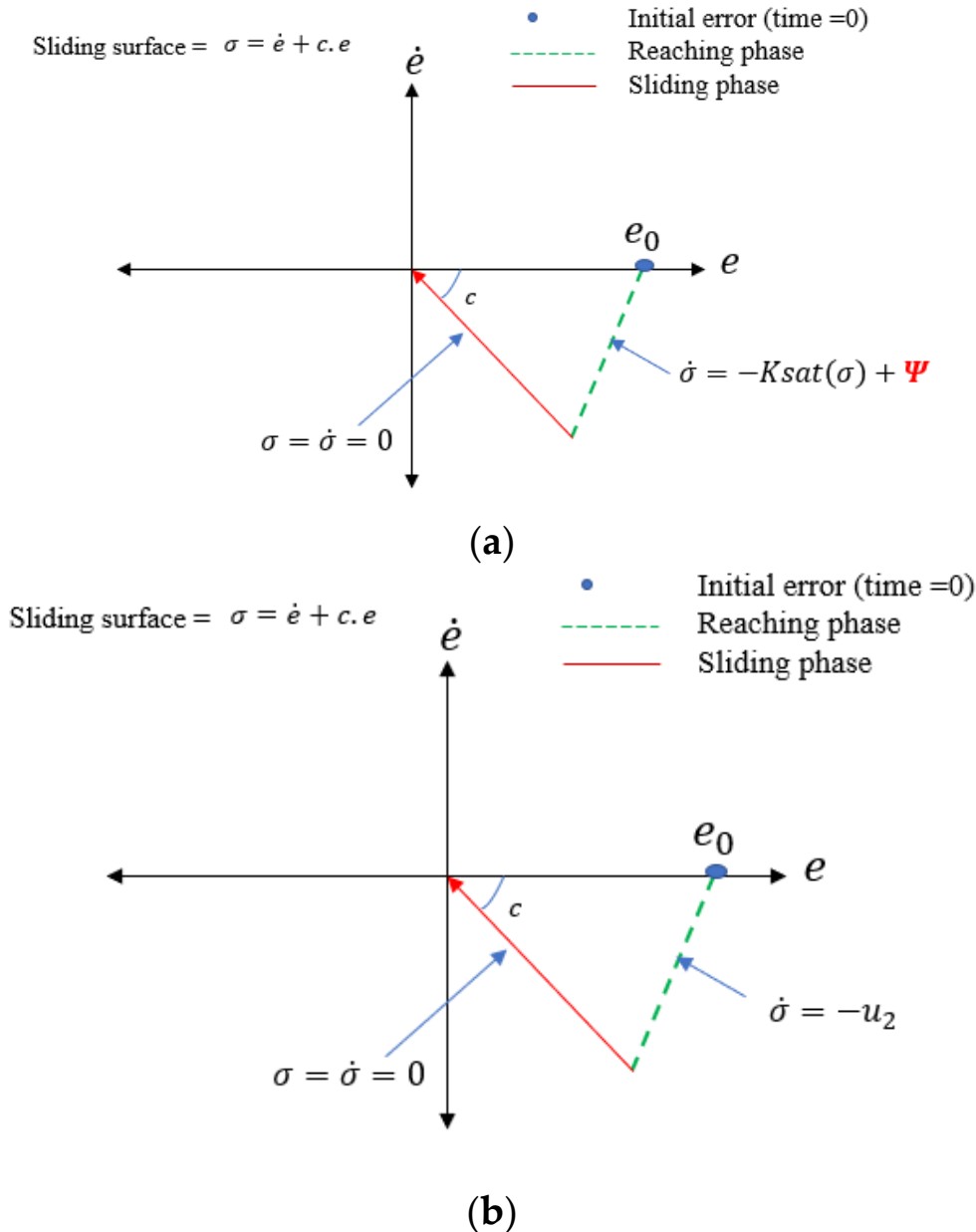

**Figure 3.** Effect of perturbation on sliding variable: (**a**) SMC (affected by perturbation), (**b**) ISMC (perturbation free).

3.1.2. Identification of Linear Parameters to Apply SMC

The control input of SMC can be observed in (10), where the control input requires the information about the dynamics of the plant ($f(x, \dot{x}, t)$). This is described in the human heart rate modeling part, which is limited as approximately linearized dynamics. The identification of linearized dynamics requires experiments to determine the equivalent impulse response and further processing to achieve better estimation of parameters through a cross-correlation coefficient, which is time consuming. The controller design requires a nominal model of the system ($f(x, \dot{x}, t)$), as shown in Figure 4. It is a challenge to extract the nominal model of the system. Therefore, ISMC was designed in this study because the controller output does not require information about system dynamics.

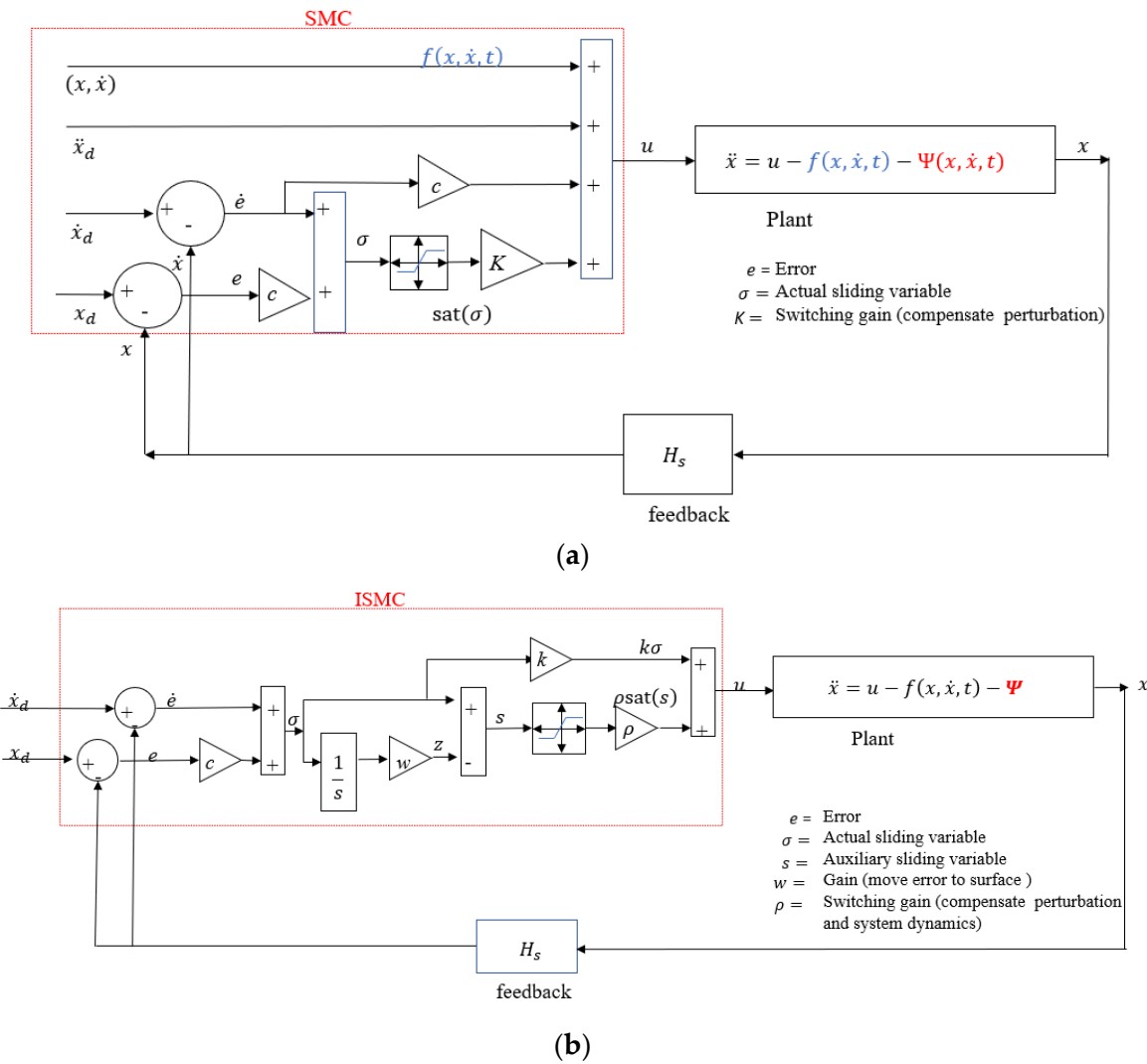

**Figure 4.** Detailed block diagram of proposed control algorithms: (**a**) SMC, (**b**) ISMC.

*3.2. Integral Sliding Mode Control (ISMC)*

To overcome the mentioned problems, ISMC was used for the desired heart rate tracking. The main idea was to design the auxiliary sliding surface ($s$), which always remains at zero ($s = 0$) [36–38]. The auxiliary sliding surface can be presented as:

$$s = \sigma - z, \tag{18}$$

$$z = -\int w{\cdot}\sigma, \tag{19}$$

where $\sigma$ is the actual sliding surface and $w$ is the positive gain. Initial condition (20) should be enforced in order to keep auxiliary sliding surface at zero ($s = 0$):

$$s_0 = \sigma_0 - z_0 = 0, \tag{20}$$

$$\sigma_0 = z_0. \tag{21}$$

In ISMC, the control input consists of two different parts:

$$u = u_1 + u_2, \tag{22}$$

$$u_1 = \rho\,\mathrm{sat}(s), \tag{23}$$

where $\rho$ is the switching gain and sat($s$) is the switching function, defined as:

$$\text{sat}(s) = \begin{cases} s/|s|, & \text{if } |\sigma| > \varepsilon_{ci} \\ s/\varepsilon_{ci}, & \text{if } |\sigma| \le \varepsilon_{ci} \end{cases}, \tag{24}$$

where $\varepsilon_c$ is the boundary layer thickness. The second control input $u_2$ in (22) is given by:

$$u_2 = w \cdot \sigma, \tag{25}$$

where $w$ is a positive constant; $u_1$ compensates the system perturbation and system dynamics, whereas $u_2$ forces the error to converge to zero in a satisfactory time. Using (18), auxiliary sliding surface dynamics ($\dot{s}$) can be calculated as:

$$\dot{s} = -u_1 + \Psi(x, \dot{x}, t). \tag{26}$$

As can be seen, the $\dot{s}$ dynamics are affected by perturbation, and the control $u_1$ compensates for this perturbation. During the auxiliary sliding mode ($s = 0$), the equivalent control can be calculated as ($\dot{s} = 0$):

$$\dot{s} = -u_{1eq} + \Psi(x, \dot{x}, t) = 0, \tag{27}$$

$$u_{1eq} = \Psi(x, \dot{x}, t). \tag{28}$$

The actual sliding surface dynamics can be calculated as:

$$\dot{\sigma} = -u_2. \tag{29}$$

As can be observed in (29), the actual sliding surface dynamics ($\dot{\sigma}$) is perturbation-free. Therefore, the system stability is guaranteed throughout compared with SMC (17).

**Theorem 2.** *For the second-order system described by (4), ISMC always satisfies the Lyapunov stability criterion (7) during the auxiliary sliding mode ($s = 0$) as its sliding surface dynamics is free of perturbation ($\dot{\sigma} = -u_2$).*

**Proof.** It must be noted that for the system described by (4) with the proposed control law (22), the system is always stable. There is no need to fulfill any condition such as the upper bound of perturbation of (13) in ISMC. After putting the sliding surface dynamics ($\dot{\sigma} = -u + \Psi(x, \dot{x}, t)$) in (7), there is derived as

$$\sigma \cdot \dot{\sigma} \le \sigma \left( -u + \Psi(x, \dot{x}, t) \right) \le 0, \tag{30}$$

In ISMC, the control input is defined as (22). During the auxiliary sliding mode ($s = 0$), the control input $u_1$ becomes the equivalent control (28):

$$\sigma \cdot \dot{\sigma} \le \sigma \left( -\Psi(x, \dot{x}, t) - u_2 + \Psi(x, \dot{x}, t) \right) \le 0, \tag{31}$$

$u_1$ has compensated for the perturbation:

$$\sigma \cdot \dot{\sigma} \le -\sigma \cdot u_2 \le 0, \tag{32}$$

After putting (25) in (32):

$$\sigma \dot{\sigma} \le -w \cdot (\sigma)(\sigma) \le 0. \tag{33}$$

As can be observed in (33), the sliding surface dynamics is free of perturbation, so the system is stable throughout. Therefore, the performance characteristics of ISMC are better than those of the conventional SMC. The advantages of ISMC over SMC in heart rate control are presented in the following. □

### 3.2.1. Insensitive to Perturbation

As can be observed in Figure 3b, during the reaching phase ($\sigma \neq 0$), the actual sliding surface dynamics is not affected by perturbation (29). Therefore, the system always satisfies the stability condition (7). In ISMC, the system states easily converge ($\sigma = 0$) to the designed sliding surface (Figure 3b), as explained in Theorem 2. The stability of ISMC is guaranteed throughout, whereas in conventional SMC, the sliding surface dynamics is perturbed (17).

### 3.2.2. ISMC of Model-Free

As can be observed in (22), the implementation of ISMC does not require information about the plant dynamics. In human heart rate control, it is difficult and a tedious job to identify and estimate an accurate heart rate model; in addition, the estimation of the human heart rate requires long time. As can be observed in Figure 4b, the control design does not require information about the nominal model of the system. Therefore, ISMC is a feasible option to track the desired heart rate.

## 4. Simulations and Experimental Results

This section consists of two parts: first, simulation results are presented, and second, experimental results are explained in detail.

### 4.1. Simulation and Discussion

Simulation was performed using MATLAB/Simulink (Figure 5), and a second-order system was considered. A constant input was given as a reference input. Both forms of control logic, SMC and ISMC, were implemented to track the reference trajectory in the presence of disturbance as shown in Figure 6. The disturbance consists of the sum of the white noise, sine wave, and the constant terms. The heart rate model, assumed as a second-order system of the subject, was estimated using the signal compression method, which is listed in Table 3. The controller's parameters are listed in Table 4. Using relation (1), the corresponding second-order transfer function of a clinical subject X was derived as (34):

$$G_s = \frac{30.25}{s^2 + 6.6s + 30.25}. \tag{34}$$

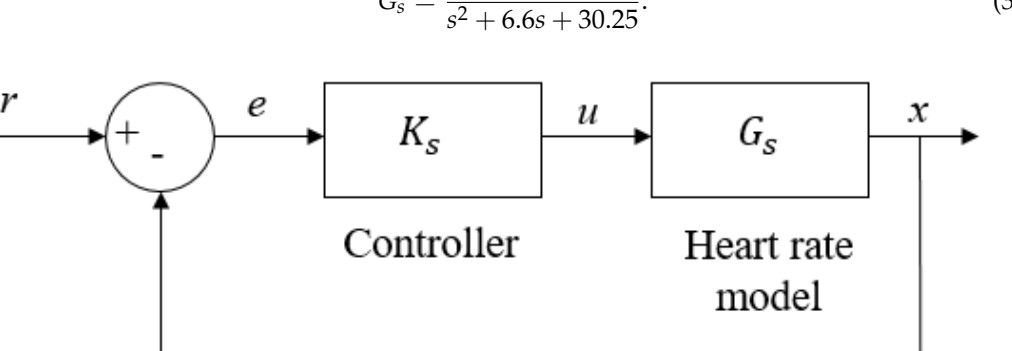

**Figure 5.** Simulink Block Diagram.

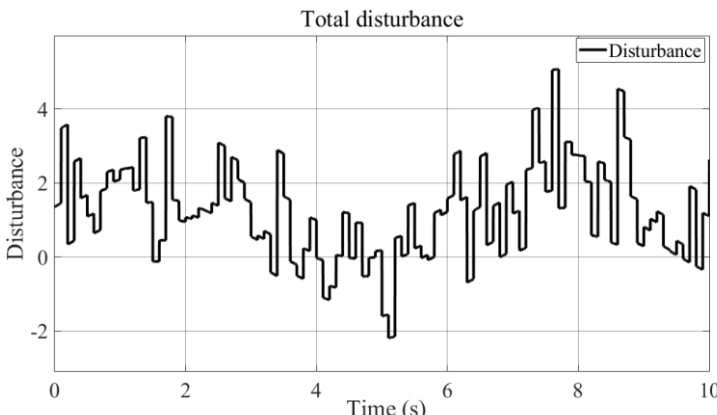

**Figure 6.** Total disturbance/perturbation.

**Table 3.** Heart model parameter of X.

| Clinical Subject | Subject Information | | | |
|---|---|---|---|---|
| | **Age** | $W_n$ | $\delta$ | **Correlation (%)** |
| X | 26 | 5.5 | 0.6 | 82 |

**Table 4.** Controller parameter for simulation.

| Number | Controller Parameter | |
|---|---|---|
| | **SMC** | **ISMC** |
| 1 | Constant input | Constant input |
| 2 | c = 5 | c = 5 |
| 3 | K = 250 | w = 20, $\rho$ = 120 |
| 4 | $\epsilon_c = 1$ | $\epsilon_{ci} = 1$ |

### 4.1.1. Better Trajectory Tracking

A constant input was given to the system in the presence of disturbance. The comparative patterns of an output response corresponding to SMC and ISMC are shown in Figure 7. ISMC output (solid blue line) tracking is better than that of conventional SMC (dotted red line). The SMC output has a fluctuation, whereas ISMC output has a smooth performance. The tracking output of SMC has a fluctuation because the actual sliding surface dynamics are affected by the perturbation (17); by comparison, the output tracking of ISMC is smooth compared with that of conventional SMC because the actual sliding surface of ISMC (29) is free of perturbation effects.

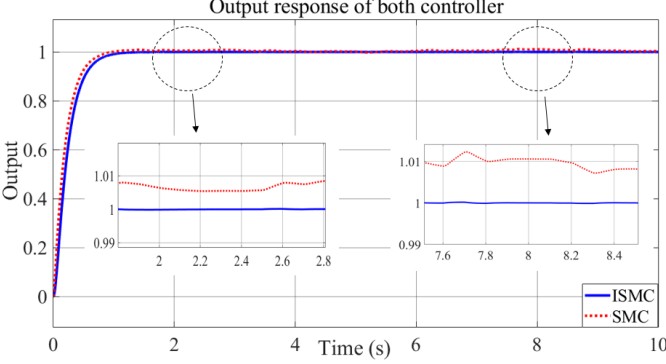

**Figure 7.** Trajectory tracking of both control schemes.

### 4.1.2. Fluctuating Sliding Surface

The sliding surface of SMC a has large fluctuation (dotted red line), as shown in Figure 8, because of the presence of the disturbance in (17). By comparison, the fluctuation in ISMC is almost negligible (solid blue line) compared with that of the SMC.

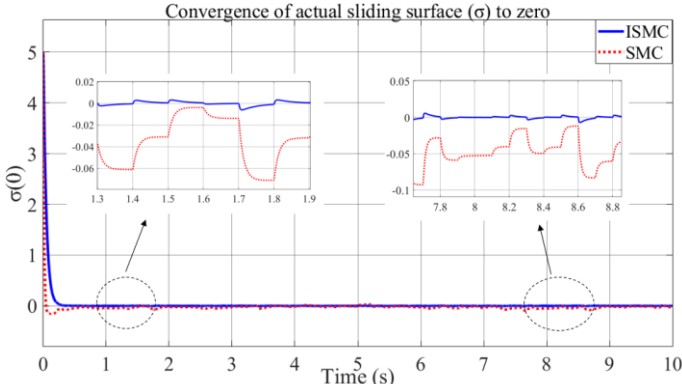

**Figure 8.** The actual sliding surface of both control schemes.

### 4.1.3. Auxiliary and Actual Sliding Surface

The auxiliary sliding variable as shown in Figure 9a always starts from zero by enforcing the initial condition (21). It was assumed that initial conditions are known:

$$\dot{x}_{d0} = 0, \qquad \dot{x}_0 = 0, x_0 = 0, x_d = 1 \tag{35}$$

$\sigma_0$ can be calculated as:

$$\sigma_0 = \dot{e}_0 + ce_0, \tag{36}$$

After putting the initial conditions (35) in (36):

$$\sigma_0 = 5, \tag{37}$$

$$\sigma_0 = z_0 = 5. \tag{38}$$

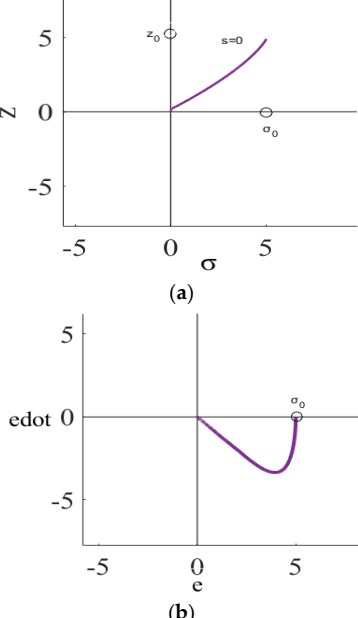

**Figure 9.** Sliding surface behavior in ISMC: (**a**) auxiliary sliding surface (s), (**b**) actual sliding surface ($\sigma$).

The response of $\sigma$, and $s$ against time (t) can be observed in Figure 10, whereas Figure 11 shows the control output of both forms of implemented logic.

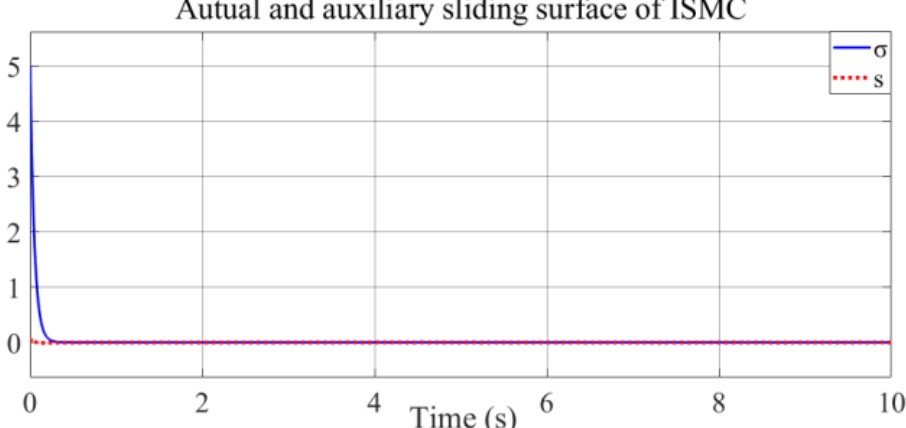

**Figure 10.** $\sigma$ converges to zero in finite time, s remains (starts from zero) at zero (ISMC).

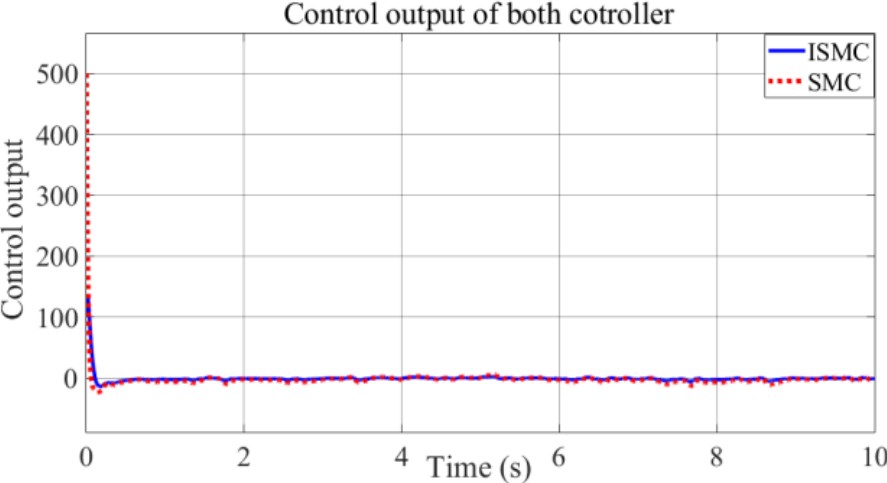

**Figure 11.** Control output of both control schemes.

### 4.2. Experimental Results

Experiments were performed to verify both controllers. The commercial treadmill, HERA 9000, is used for heart rehabilitation. The treadmill control input frequency is calibrated to find the relation between the control output command and treadmill velocity. The unit of the control output is hertz. The calibrated relation to convert hertz to velocity (Km/h) is obtained by the experiment, which was implemented in C#. The calibrated control output frequency of 45 hertz is equal to 1 km/h and 315 hertz equals 7 km/h. The maximum velocity and minimum velocity were set as 7 and 1 km/h, respectively. Figure 12 shows the controller's output in hertz (a) and the corresponding profile are conversion to velocity (b).

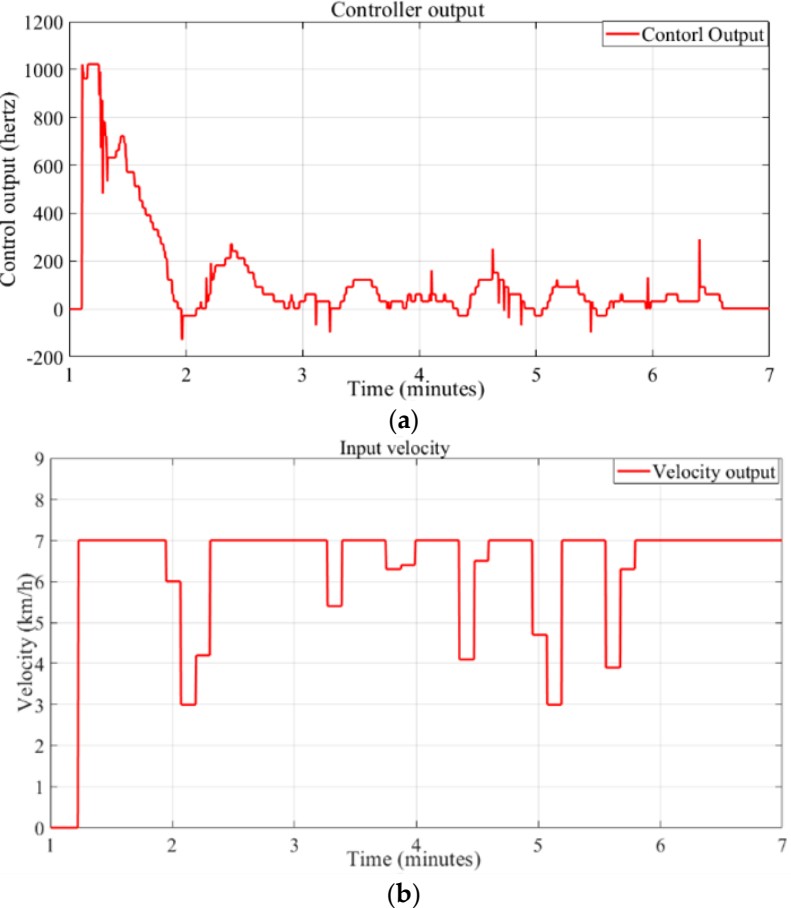

**Figure 12.** Hertz and velocity relationship: (**a**) hertz, (**b**) velocity.

The clinical subject's heart rate and treadmill velocity are visible on the monitor (GUI). There is also a stop button on the control panel of the treadmill to ensure safety in the case of an emergency, and so the observer can stop the program in the case of an emergency while running. For safety purposes, a foam mattress was placed to the rear of the treadmill during the experiment.

Experiments were conducted at Pusan National University Hospital with six clinical subjects, for which data were gathered by the hospital. Both controllers were implemented and the corresponding results were compared. The total time for the experiment was 7 min. The target heart rate was different for every clinical subject and prescribed by hospital doctors. The hospital doctors suggested the target heart rate of the subject after examining the subject's condition. Therefore, each subject has a different target heart rate. The target heart rate for the subjects is shown in Table 5.

**Table 5.** Target heart rate of each clinical subject (prescribed by a doctor).

| Clinical Subject | Subject Information | |
|:---:|:---:|:---:|
| | **Target Heart Rate (bpm)** | **Initial Heart Rate (bpm)** |
| A | 149 | 107 |
| B | 114 | 92 |
| C | 133 | 86 |
| D | 120 | 104 |
| E | 119 | 73 |
| F | 119 | 95 |

Firstly, SMC was used to track the desired heart rate trajectory for 7 min. After taking a rest during a 10 min break, the second controller (ISMC) was implemented for the same task and period of time. The heart rate tracking error using SMC for the six different subjects is shown in Figure 13a. It is known that the error trajectory of each subject lies between plus 10 bpm and minus five during a steady state. This steady-state error is large. To decrease this error, ISMC was implemented in the next experiments.

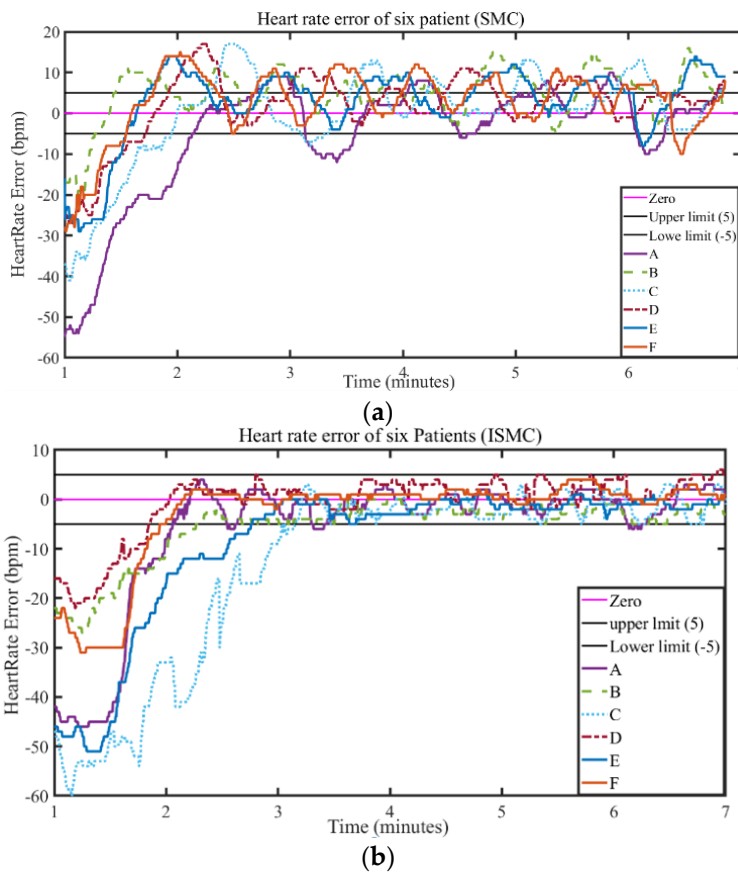

**Figure 13.** Heart rate tracking error of six subjects: (**a**) SMC, (**b**) ISMC.

In the second experiment, ISMC logic was implemented. The results are shown in Figure 13b, and show that the steady-state error of ISMC is mostly between ±5 (bpm), which is smaller than that of SMC. This research confirmed that ISMC can reduce the steady-state error by more than SMC. This means that ISMC can reduce the tracking error even if ISMC does not require a mathematical model of the human heart rate. However, the control output of ISMC has a little chattering, which should be reduced in the future. ISMC will be integrated with the observer for disturbance rejection. A sliding perturbation observer (SPO) or nonlinear extended state observer (ESO) will be integrated with ISMC for this purpose. The disturbance observer-based ISMC will further improve system performances, such as by reducing chattering.

## 5. Conclusions

The human heart is a nonlinear system, and accurate heart rate modeling is a tedious task. To track the desired heart rate during treadmill exercise, a robust controller is required. Therefore, in this research, two different nonlinear controllers were designed and implemented on the system. In SMC heart rate tracking, a large steady-state error was observed because of the effect of the assumed perturbation on the sliding surface. In addition, the control input of SMC required a plant dynamics model with linear parameters, which are difficult to model. Therefore, ISMC was implemented to reduce tracking errors under

unknown dynamic model. In ISMC, the effect of the perturbation on the actual sliding surface was eliminated by the design of an auxiliary sliding surface; furthermore, the control input of ISMC does not require a system mathematical model. This ISMC showed outstanding performance, including less steady-state error when compared with SMC in both simulation and experimental results. However, the control output of ISMC has a little chattering, which is not desirable. In the future, this ISMC will integrate with a sliding perturbation observer (SPO) to reduce the chattering.

**Author Contributions:** Conceptualization, S.J.A. and M.C.L.; methodology; M.C.L., B.J.L. and M.J.S.; Software, W.J.K.; validation, S.J.A., M.C.L., J.K., B.J.L. and M.J.S.; writing, S.J.A., W.J.K. and M.C.L.; project administration, M.C.L., B.J.L. and M.J.S.; funding acquisition, M.C.L., B.J.L. and M.J.S. All authors have read and agreed to the published version of the manuscript.

**Funding:** Korea Institute for Advancement of Technology (KIAT)(P0008473): C; Korea Institute of Evaluation and Planning and the Ministry of Trade, Industry & Energy(MOTIE) of the Republic of Korea: 20214000000410.

**Data Availability Statement:** Not applicable.

**Acknowledgments:** This research was supported by Korean Institute for Advancement of Technology (KIAT) grant funded by the Korea Government (MOTIE) (P0008473), and also supported by Korea Institute of Evaluation and Planning (KETEP) and the Ministry of Trade, Industry & Energy (MOTIE) of the Republic of Korea (Number: 20214000000410).

**Conflicts of Interest:** The authors declare no conflict of interest.

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
