# Peer review of "Robust Control Design of a Human Heart Rate System for Cardiac Rehabilitation Exercise"

_electronics, doi:10.3390/electronics11244081_

Round 1

Reviewer 1 Report

This manuscript reports the tracking of heart rate using two different nonlinear controllers. It turned out that ISMC could perform better in terms of a low steady-state error. The reviewer is looking forward to seeing a more matured technology after addressing the chattering problem. I would recommend the publication of the work after the authors address the following comments.

1.      The third paragraph describes the heart model. Two very recent papers should be cited there to enrich the development of the field. They are “H. Chang, Q. Liu, J.F. Zimmerman, K.Y. Lee, Q. Jin, M.M. Peters, M. Rosnach, S. Choi, S.L. Kim, H.A.M. Ardoña, et al. Recreating the heart’s helical structure-function relationship with focused rotary jet spinning. Science, 377 (2022), pp. 180-185, 10.1126/science.abl6395.” and “M. Liu, et al. Focused rotary jet spinning: A novel fiber technology for heart biofabrication. Matter. Volume 5, Issue 11, 2 November 2022, Pages 3576-3579.”

2.      The quality of some figures needs to be improved. For example, figure1 is distorted, and the scale bar of figure 1b and c is missing.

Author Response

Answer Sheet to Reviewer 1’s Comments

Thank you very much for your valuable comments. Your suggestions have improved the quality of the manuscript.  The authors have tried to answer as follows and revised the manuscript which is highlighted in yellow color in the revised manuscript.  

The answer sheets are attached as PDF file. 

Reviewer 2 Report

1. The content format and English expression of the manuscript should be modified carefully before resubmission.

For example:

As system is nonlinear therefore there is a need of robust control to reduce output error. (Line 74);

SMC has two phases reaching phase and the sliding phase (Line 80);

In ISMC auxiliary sliding variable (line 93); I

SMC design (Line 96); Hear rate (Line 158);

The second controller ISMC had implemented (Line 477), etc.

2. The image of Fig. 3 is unclear, please update the diagram, and there is no illustration and explanation of Fig. 3 to state the difference of the result. Meanwhile, please also interconnect the Fig. 4 to corresponding illustration for better understanding.

3. The equation about the second order system is different from that in your reference 13, could you explain it?

4. Could you explain why ISMC can achieve better non-linear simulation compared to SMC by derived equation?

5. Could you provide the basis for converting from 25 hertz to 1 km/h in the 396th line?

6. Could you provide the references or proof to support the idea ’the integration of SPO or ESO can promote the ISMC’?

7. In the Fig. 13, the fluctuations in curves of ISMC is more obvious than those in SMC in the initial stage, could you explain the results? And will those perturbations effect the monitoring result? Moreover, please use the same value range for better comparison.

Author Response

Thank you very much for your valuable comments. Your suggestions have improved the quality of the manuscript.  The authors have tried to answer as follows and revised the manuscript which is highlighted in yellow color in the revised manuscript.  

The answer sheets are attached as PDF file.

Round 2

Reviewer 2 Report

All of my comments have been well addressed. The manuscript can be accepted in its current form.